# Community size rather than grammatical complexity better predicts Large Language Model accuracy in a novel Wug Test

**Nikoleta Pantelidou**[1☉*], **Evelina Leivada**[1,2☉], **Raquel Montero**[1], **Paolo Morosi**[1*]

**1** Departament de Filologia Catalana, Centre de Lingüística Teòrica, Universitat Autònoma de Barcelona, Bellaterra, Barcelona, Spain, **2** Institució Catalana de Recerca i Estudis Avançats (ICREA), Barcelona, Spain

☉ These authors contributed equally to this work.
* nikoleta.pantelidou@autonoma.cat (NP); paolo.morosi@uab.cat (PM)

## Abstract

The linguistic abilities of Large Language Models are a matter of ongoing debate. This study contributes to this discussion by investigating model performance in a morphological generalization task that involves novel words. Using a multilingual adaptation of the Wug Test, six models were tested across four partially unrelated languages (Catalan, English, Greek, and Spanish) and compared with human speakers. The aim is to determine whether model accuracy approximates human competence and whether it is shaped primarily by linguistic complexity or by the size of the linguistic community, which affects the quantity of available training data. Consistent with previous research, the results show that the models are able to generalize morphological processes to unseen words with human-like accuracy. However, accuracy patterns align more closely with community size and data availability than with structural complexity, refining earlier claims in the literature. In particular, languages with larger speaker communities and stronger digital representation, such as Spanish and English, revealed higher accuracy than less-resourced ones like Catalan and Greek. Overall, our findings suggest that model behavior is mainly driven by the richness of linguistic resources rather than by sensitivity to grammatical complexity, reflecting a form of performance that resembles human linguistic competence only superficially.

## Introduction

Large Language Models (LLMs) are Artificial Intelligence systems designed to interact using human language. Their high performance across different domains, including education, medicine, finance, and translation [1] stems from their ability to generate contextually appropriate and syntactically diverse responses, which in turn reflects a sophisticated manipulation of linguistic structures and rules [2]. Despite these achievements, however, the linguistic abilities of LLMs remain a matter of

**Data availability statement:** All data files are available from the OSF database (https://osf.io/4z5n6/).

**Funding:** EL acknowledges funding from the Spanish Ministry of Science, Innovation & Universities MCIN/AEI/https://doi.org/10.13039/501100011033) under the research project CNS2023-144415. The funders had no role in study design, data collection and analysis, decision to publish, or preparation of the manuscript.

**Competing interests:** The authors have declared that no competing interests exist.

ongoing debate. It has been observed, for instance, that the mechanisms through which LLMs learn and process language differ fundamentally from those underlying human cognition [3]. In the semantic domain, it has been argued that signifiers are not accessible to LLMs [4] and that models lack representations of words comparable to those in the human mind [5]. Consequently, LLMs are often described as possessing only functional competence, but lacking conceptual meaning [4]. In addition, several studies report that models underperform compared to humans in tasks requiring grammaticality judgments, both in terms of accuracy and consistency [6–10]. It thus remains to be established whether the functional competence of LLMs not only approaches that of humans but also extends to novel material not included in their training data [11].

This question is particularly relevant because LLMs are trained on vast amounts of data from the internet, whose quantity and quality are crucial determinants of their performance [1,12,13]. If the data are not carefully selected and preprocessed, models may produce biased, harmful, and stereotypical responses [14–16]. Since such biases are present in all textual sources, companies developing LLMs attempt to mitigate them through the choice of resources and subsequent filtering and evaluation procedures [16].

Against this backdrop, several studies have investigated the extent to which LLMs can manipulate different languages in ways that demonstrate human-like abilities and an understanding of underlying linguistic rules, even when confronted with inputs absent from their training data. The existing literature, however, has focused predominantly on the syntactic and semantic abilities of LLMs, with relatively little attention to morphology – the component of language that generates words or lexemes according to systematic patterns of covariation in form and meaning [17]. Two relevant studies nonetheless examine the morphological capacities of LLMs, using multilingual adaptations of the Wug Test [18]. The Wug Test was originally designed to assess whether children apply grammatical rules to novel words, by asking participants to provide an inflected or derived form of a nonce word. In the original paradigm, participants were introduced to a fictional character with a sentence such as "This is a wug.", containing the nonce word *wug*. They were then shown two of these characters and prompted to complete the sentence "Now there are two ___.". The target answer, *wugs*, indicated that participants possessed an internal representation of English pluralization rule, extending beyond rote memorization.

In the context of LLMs, previous work has assessed the morphological capabilities of ChatGPT-3.5 through a multilingual adaptation of the Wug Test [19]. Their experiment used invented words in English, German, Tamil, and Turkish, to evaluate the model's ability to generalize morphological rules – particularly, plural formation – to unseen data. The findings revealed that, although never reaching the performance of the best human annotator or the strongest baselines, ChatGPT-3.5 performed best in German, surpassing English, Turkish, and Tamil. This outcome is intriguing given that English exhibits a simpler morphological system than German in the nominal domain, typically involving the suffix *-s* or *-es* [20], with only a limited number of irregular forms. German, by contrast, employs a variety of pluralization strategies, including

several suffix classes (e.g., -e, er, -n/en, -s) and frequent stem modifications (e.g., umlaut), which collectively contribute to a high degree of morphological complexity [21]. [19] therefore suggest that factors beyond morphological complexity must have influenced the model's generalizations. At the same time, however, since English is far more represented in the training data, the findings also imply that multiple proxies – including but not limited to data exposure – likely shaped the model's performance. Two related questions follow from this study: (i) does morphological complexity influence the performance of LMMs on morphological tasks such as the Wug Test? And (ii) does the interaction between linguistic complexity and language-community size affect LLMs' performance across languages?

A partial response comes from [22], who ran the Wug Test in French, German, Portuguese, Romanian, Spanish, and Vietnamese with both ChatGPT-3.5 and ChatGPT-4. In their study, the original English Wug Test was translated into the respective languages, and linguistically trained native speakers evaluated the translations. Their results show that both models generally succeeded in generating the target morphemes for nonce words, with GPT-4 slightly outperforming GPT-3.5. More broadly, [22] argue that LLMs' success in generating correct forms is predicted by the language's morphological complexity, particularly *integrative* complexity, which refers to the degree of predictability of inflected forms. A summary of the most relevant studies is given in Table 1.

Given the contrasting findings of [19] and [22], several questions remain unresolved. Concretely, it is unclear whether LLM performance in morphological tasks is primarily driven by linguistic complexity or by the size of the language community and its representation in training data. Furthermore, the scope of existing work is limited: [19] tested only ChatGPT-3.5, while [22] compared ChatGPT-3.5 and ChatGPT-4, leaving the overall picture fragmented and model-specific. As a result, we lack a comprehensive account of how LLMs handle morphological generalization across languages. The present study addresses this gap by systematically testing six models (ChatGPT-3.4, ChatGPT-4, Grok 3, Bert, DeepSeek and Mistral) across four partially unrelated languages (Catalan, English, Greek, and Spanish), comparing their performance with human speakers through a multilingual adaptation of the Wug Test.

## On linguistic complexity and community size

Discovering what modulates the linguistic abilities of LLMs is paramount for understanding what model scaling can do, and whether it can lead to better linguistic performance. Based on previous literature [19,22], the two important concepts behind our study design are *linguistic complexity* and *community size*. Linguistic complexity is a multifaceted notion whose definition varies across different subfields and can be identified with structural, cognitive, and developmental complexity [23]. Since the present study focuses on LLM morphological performance, the relevant dimension is that of structural complexity, understood as the quantity of overt formal features in a given language and the way in which they are organized and interconnected.

With respect to morphology, various measures of complexity have been proposed. For verbal morphology, for instance, the frequency of tensed forms, the variety of past tense structures, and the number of various verb inflections are often considered [24]. The present study, however, focuses exclusively on nominal inflection, and in particular plural formation.

**Table 1. Summary of previous studies.**

| Study | Title | Key finding |
|-------|-------|-------------|
| Dang et al. 2024a | Morphology Matters: Probing the Cross-linguistic Morphological Generalization Abilities of Large Language Models through a Wug Test | The amount of training data is more important than a language's morphological complexity |
| Dang et al. 2024b | Tokenization and Morphology in Multilingual Language Models: A Comparative Analysis of mT5 and ByT5 | Languages with many irregularities benefit more since they have a larger presence in the training data |
| Weissweiler et al. 2023 | Counting the Bugs in ChatGPT's Wugs: A Multilingual Investigation into the Morphological Capabilities of a Large Language Model | The influence of factors beyond linguistic complexity on the model's morphological generalization should be considered. |

Therefore, to operationalize morphological complexity across the four languages under investigation, we adopted the method proposed by [25]. This approach distinguishes two dimensions of morphological complexity: *fusion* and *informativity*. The fusion dimension captures the extent to which a given language uses phonologically bound markers (i.e., affixes) rather than phonologically independent markers. Languages with affixes encoding tense, aspect, and mood on verbs, or case, gender, and number on nouns and pronouns, receive higher fusion scores. The informativity dimension, by contrast, reflects the number of obligatory grammatical distinctions marked in a language: the more categories obligatorily expressed, the higher the informativity score.

To calculate the fusion and informativity scores in the languages we test, we employ Grambank v1.0 [26], a global database covering 2,467 languages and coding 195 structural features relevant to fusion and informativity. Complexity was calculated as a global measure, following [25]'s procedure. Using Python [27] in the Spyder environment [28], we computed scores for each language as follows. For fusion, only features with a Fusion weight of 1 were considered: each language received one point if the feature was present (coded as 1) and zero points if absent (coded as 0). The Fusion score for each language was calculated as the mean of these features. Features with Fusion weights of 0, 0.5, or missing values were excluded. For informativity, features were grouped by grammatical function (e.g., singular, tense). A language was counted as marking a function if at least one feature in the group was coded as present, and the Informativity score was calculated as the proportion of marked groups relative to the total number of groups with at least one present feature. The code, input files, and results are available at https://osf.io/4z5n6/.

The results show clear differences across languages (Table 2). English displays the lowest fusion score (0.29), while Greek scores highest (0.53). Greek also ranks highest in informativity (0.44), closely followed by Spanish (0.42), with English again lowest. Catalan and Spanish show very similar values on both metrics, though Catalan's average score (0.4165) is slightly higher than Spanish's (0.4125). Ordering the languages from least to most complex yields: English < Spanish < Catalan < Greek.

Turning to community size and LLMs' training data, it is important to note that English, spoken by hundreds of millions of native speakers and used globally as a lingua franca, overwhelmingly dominates the online domain, resulting in a vast representation in the training corpora. Spanish, with a large global community of native and second-language speakers, also benefits from an abundant digital footprint, though still smaller in scale than English. By contrast, languages with smaller speaker populations, such as Greek or Catalan, are usually represented less extensively. That said, the correlation between population size and training data availability is not strictly linear. Catalan, for example, has fewer speakers than Greek, but benefits from a relatively strong digital infrastructure thanks to cultural and political initiatives promoting its use online. Conversely, widely spoken languages with large populations but less online visibility—such as Hindi or Bengali—remain underrepresented relative to their number of speakers. Thus, while larger speaker communities generally increase the likelihood of richer training data, factors such as digitization policies, cultural prestige, and technological adoption also play a decisive role.

For the purposes of this study, and in the absence of precise information regarding the exact amount of training data used by each model, we make the plain assumption that community size directly correlates with the amount of training data available to the models. Accordingly, the ordering of tested languages by community size/training data is: English (~1.5 billion speakers) > Spanish (~488 million speakers) > Greek (~12 million speakers) > Catalan (~8 million speakers).

**Table 2. Fusion and informativity scores per language.**

| Language_ID | Language | Fusion_scores | Informativity_scores | Mean scores |
|---|---|---|---|---|
| mode1248 | Greek | 0.538462 | 0.448980 | 0.493721 |
| stan1289 | Catalan | 0.418182 | 0.423077 | 0.4206295 |
| stan1288 | Spanish | 0.400000 | 0.425926 | 0.412963 |
| stan1293 | English | 0.291667 | 0.285714 | 0.2886905 |

Finally, it is also worth noting that the traditional view often assumes a close correlation between linguistic complexity and community size. According to the linguistic niche hypothesis [29,30], the sociolinguistic environment shapes linguistic complexity: small, homogeneous communities with mostly native speakers (i.e., esoteric communities) tend to preserve or develop greater morphological complexity, whereas large, heterogeneous communities with many L2 learners (i.e., exoteric communities) tend toward simplification. This relationship, however, remains debated. [31] indeed highlighted the role of non-native speakers, arguing that exoteric communities with high proportions of L2 speakers tend toward simplification, while esoteric communities preserve irregularities. [32] provided further evidence in support of this view, showing that larger societies may evolve simpler grammatical systems. By contrast, [33] found that it is the absolute number of speakers, rather than the proportion of L2 speakers, that correlates with complexity. More recently, [25] reported only a very weak correlation between population size and (reduced) complexity. Taken together, these findings suggest that while community size can affect linguistic complexity, the effect is neither straightforward nor uniform but shaped by additional sociolinguistic and demographic factors. This debate makes it particularly relevant to examine the interplay between linguistic complexity and community size in the context of LLM performance across languages.

**The present study**

This study investigates how LLMs extend morphological generalizations across languages, with particular attention to the impact of linguistic complexity and community size. To frame this investigation, we articulate three guiding research questions (RQs):

• RQ1: Do LLMs exhibit human-like behavior in the generalization of novel morphological forms, or do they deviate from human baselines?

• RQ2: Does the linguistic complexity of a language influence model performance? If so, to what extent?

• RQ3: Alternatively, is model accuracy primarily conditioned by the amount of training data and the size of the speaker community?

To address these questions, we designed a multilingual adaptation of the Wug Test, systematically evaluating six models (ChatGPT-3.5 ([34]), ChatGPT-4 ([35]), Grok 3 ([36]), BERT ([37]), DeepSeek ([38]), and Mistral ([39])) across four partially unrelated languages (Catalan, English, Greek, and Spanish), and comparing their performance with the responses of human speakers.

The RQs give rise to three testable predictions. With respect to RQ1, prior work suggests that LLMs perform reliably in tasks that involve relatively constrained morphological operations (e.g., [19,22]). Accordingly, we expect models to approximate human behavior on a task as elementary yet revealing as the Wug Test.

For RQ2, if structural complexity exerts a decisive influence, then LLMs' should perform better in languages with lower complexity. Specifically, their performance is expected to follow the ranking:

English > Spanish > Catalan > Greek.

Regarding RQ3, if the main factor influencing LLMs' performance is community size, and, by extension, the amount of training data, we instead expect the ranking to be: English > Spanish > Greek > Catalan.

These predictions were evaluated by directly comparing model and human performance across the four tested languages. The analysis focuses on overall accuracy rates, cross-linguistic patterns, and the interaction between linguistic complexity and data availability. By examining where and how models diverge from human baselines, the study aims to determine whether their behavior reflects genuine morphological competence or merely sensitivity to distributional properties in the input. This approach also allows us to assess which of the two factors (i.e., structural complexity or resource availability) better accounts for performance differences across languages.

## Methodology

The task employed in this study is a modified version of the Wug Test [18]. The Wug Test assesses the ability to apply grammatical rules pertinent to inflectional morphology to words that participants have never encountered before. As mentioned in the Introduction, the choice of this test is motivated by previous literature [[19,22]], which provided interesting results regarding the morphological abilities of LLMs in this task, but also generated questions regarding what drives their performance. Specifically, we designed a novel version of the Wug Test featuring 30 test items: 15 words consisting of two syllables and 15 words of three syllables. The full task is available in the OSF repository (https://osf.io/4z5n6/).

The task was created and hosted on the PCIbex Farm platform ([40]). The sentences were presented in the written modality on the screen, and participants were asked to type their responses in the designated blank spaces using a computer or a mobile device. The primary objective of the task was to elicit the plural form of each nonce word. To minimize confounds, each test item followed an identical structure, differing only in the novel word presented, as exemplified below.

**[ENG]** *Continue the phrase with only one word. Here is an example: Yesterday a glorp appeared in my garden. Today there was another one. Now there are two? [response: glorps].*

*Continue the phrase with only one word: Yesterday a sottle appeared in my garden. Today there was another one. Now there are two? ______*

This uniformity ensured that neither semantic properties nor contextual cues could influence responses, thereby allowing a direct comparison between human participants and language models under equivalent linguistic conditions. The test was developed in 4 languages differing in size and levels of complexity, namely Catalan, English, Greek, and Spanish, and evaluated by native speakers of each language.

The nonce words created in this study respect the morphophonological constraints of each target language. They were systematically derived from existing lexical items in each target language by altering the initial consonant. This design choice was motivated by evidence showing that consonants play a more critical role than vowels in lexical identification and word recognition for both humans and LLMs [41]. Consequently, manipulating consonants provides more robust and reliable stimuli. In addition, the novel words were balanced within each language according to syllable count and grammatical gender. This control minimized the risk of inadvertent biases toward particular phonological patterns or gendered forms and ensured the cross-linguistic comparability of the stimuli. Table 3 provides sample items from each language, illustrating the distribution of two- and three-syllable words and gender marking where applicable.

## Participants

A total of 160 participants (78 F) took part in the study, with 40 adult native speakers per language to ensure balanced representation across the tested languages. Participants were recruited via the online platform Prolific and compensated for their participation. All participants provided written informed consent before taking part in the study. The experiment was carried out in accordance with the Declaration of Helsinki and had the written approval of the ethics committee (*Comité d'Ètica en la Recerca* (CERec)) of the Autonomous University of Barcelona (application no. 7150). The

**Table 3. Sample of the stimuli per tested language.**

| English | | Spanish | | Catalan | | Greek | |
|---|---|---|---|---|---|---|---|
| 2-syllable | 3-syllable | 2-syllable | 3-syllable | 2-syllable | 3-syllable | 2-syllable | 3-syllable |
| jater | mucumber | danta (FEM) | sestino (MASC) | gavall (MASC) | frúixola (FEM) | λέφα (FEM) | λύννεφο (NEU) |
| bocket | rospital | ñafa (FEM) | meclado (MASC) | famí (MASC) | zoquina (FEM) | τάμπα (FEM) | ζεβύρι (NEU) |
| capkin | forpedo | zulta (FEM) | fepillo (MASC) | deó (MASC) | flimona (FEM) | φέστη (FEM) | ρεχνίδι (NEU) |

recruitment started on March 15th, 2025, and ended on May 15th, 2025. Exclusion criteria included self-reported cognitive, neurological, hearing, or speech-related impairments. In addition, participants who failed to complete at least 50% of the task in the target way were removed from the final sample. No time limits were imposed on task completion, although participants were required to respond to all prompts in order to finish the experiment.

The same test was also administered to six LLMs: ChatGPT-3.5, ChatGPT-4, Grok 3, BERT, DeepSeek, and Mistral. Model evaluation was primarily conducted manually through the respective user interfaces, with the exception of the BERT model, which was evaluated programmatically using Python due to its comparatively slow response time when accessed manually. For the manual evaluation, each prompt was entered independently into a new chat session without any prior conversational context. After receiving the model's response, the conversation history was deleted to avoid contextual bias across prompts.

Since no paid subscriptions were used for any of the evaluated models, usage was constrained by the limitations imposed on free-tier access. Consequently, testing was periodically interrupted upon reaching usage limits, requiring waiting periods before further evaluation could proceed. As a result, the manual testing process extended over approximately one month. In contrast, the automated evaluation of the BERT model, conducted via Python, was completed in under two hours, highlighting a substantial difference in efficiency between manual and automated testing procedures.

Regarding computational resources, BERT was free. The implementation relied on standard open-source libraries relevant to natural language processing, including AutoTokenizer and AutoModelForMaskedLM from the Hugging Face [42]. No specialized hardware (e.g., GPUs or TPUs) was employed, and all computations were performed on general-purpose computing resources. The raw datasets, the code used for the analyses, and the experimental stimuli are available in the OSF repository. Fig 1 diagrams the experimental design and testing process.

## Data annotation

The central measure of interest of the present study was accuracy in the pluralization of nonce words across human participants and models, as the aim was examining possible effects of agent (i.e., humans vs. LLMs), language complexity, and community size on accuracy. Accuracy was measured by comparing each response with a predefined target based on the morphophonological rules of the relevant language. Correct responses were scored as 1 and incorrect responses as 0. Both stressed and unstressed forms were accepted, provided that the stress placement was accurate. Misplaced stress, in contrast, considered a violation of the phonological rules of the language, were coded as inaccurate. With respect to pluralization strategies, all test items conformed to regular patterns based on the morphological rules of each language. Misspelled answers or misapplied irregular forms were also scored as inaccurate. The same coding procedures were applied to human and model responses.

## Results and data analysis

Turning first to human participants, descriptive analyses revealed some cross-linguistic variation. English speakers achieved the lowest accuracy (84.8%), followed by Catalan (90.9%), Greek (94.8%), and Spanish (95.7%). These differences are shown in Fig 2.

In order to test if these differences were statistically significant, we conducted in R [43] (version 4.5.1) a mixed-effects logistic regression model (glmer) [44] with Accuray (correct, incorrect) as the dependent variable and Language (Catalan, English, Greek and Spanish) as the independent variable. Participants and Items were added as crossed-random factors. The null model (accuracy ~ 1 + (1| item) + (1| participant)) was compared with the model including Language as a fixed effect (accuracy ~ Language +(1| item) + (1| participant)) via the anova()-function. Including Language significantly improved model fit: $p < 3.1e\text{-}08$. Additionally, random slopes for Language were included within the items as this also significantly improved model fit (as estimated by comparing the logLikelihoods of the models using the anova()-function). Results showed a main effect of Language ($\chi^2 = 24.792$, $p < 0.001$) on accuracy. Post-hoc comparisons were then run

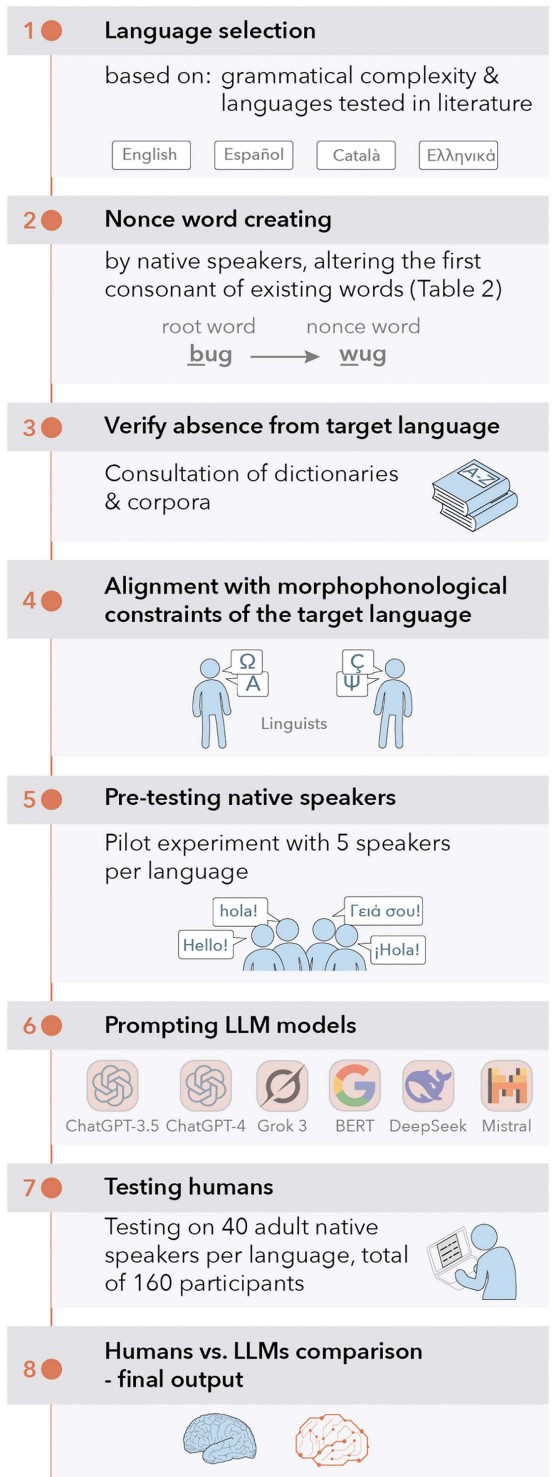

**Fig 1. Experimental design and testing process.**

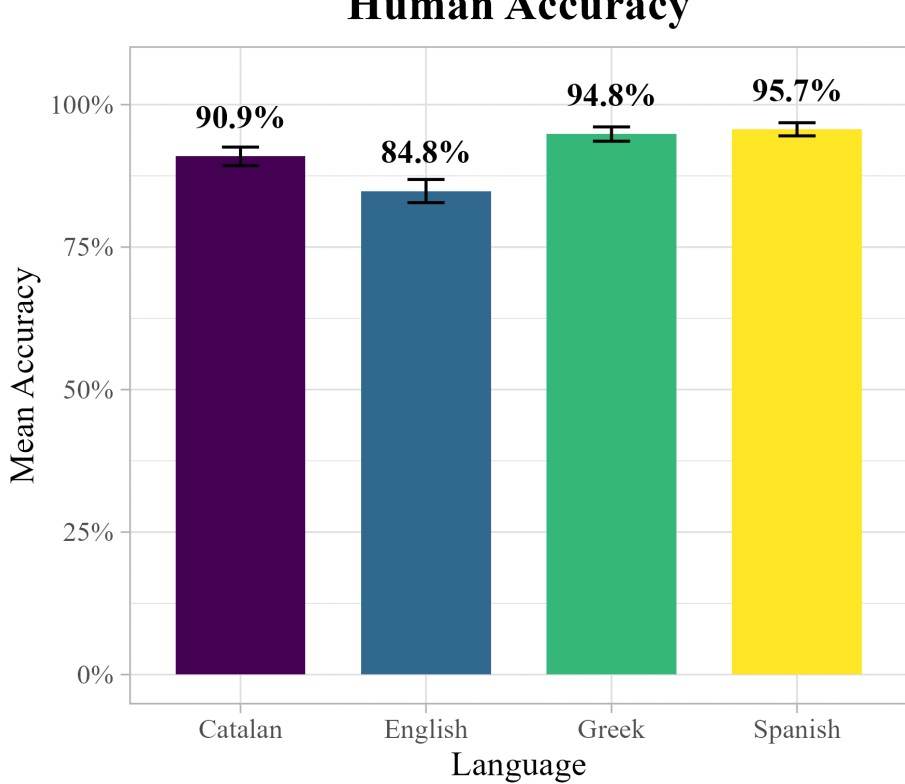

**Fig 2. Humans' performance across languages in the Wug Test.**

using the emmeans()-function [45] with Tukey adjustment. Table 4 shows the results obtained. While English was significantly different from the other languages, no significant differences were found across the other languages. These results indicate that, even in humans, morphological generalizations show language-dependent variation, with English pluralization posing greater challenges. As we discuss in the Discussion section, the difficulty of specific test items may have contributed to English being ranked lowest.

Direct comparisons between humans and models showed broadly similar levels of performance, except for the BERT model, whose performance was at floor across all languages (Fig 3). In English and Spanish, models generally outperformed humans, with ChatGPT-4, DeepSeek, Grok 3, and Mistral reaching 100% accuracy in Spanish, whereas humans

**Table 4. Results of the post-hoc test on human responses. Values are given in log-odds ratio. P-value adjustment was conducted with the Tukey method.**

| Contrast | ß | SE | z | p |
|---|---|---|---|---|
| Catalan – English | 1.671 | 0.574 | 2.913 | 0.0188* |
| Catalan – Greek | 0.145 | 0.598 | 0.243 | 0.9950 |
| Catalan – Spanish | −0.655 | 0.724 | −0.904 | 0.8027 |
| English – Greek | −1.526 | 0.443 | −3.441 | 0.0032* |
| English – Spanish | −2.326 | 0.560 | −4.151 | 0.0002** |
| Greek – Spanish | −0.800 | 0.593 | −1.348 | 0.5322 |

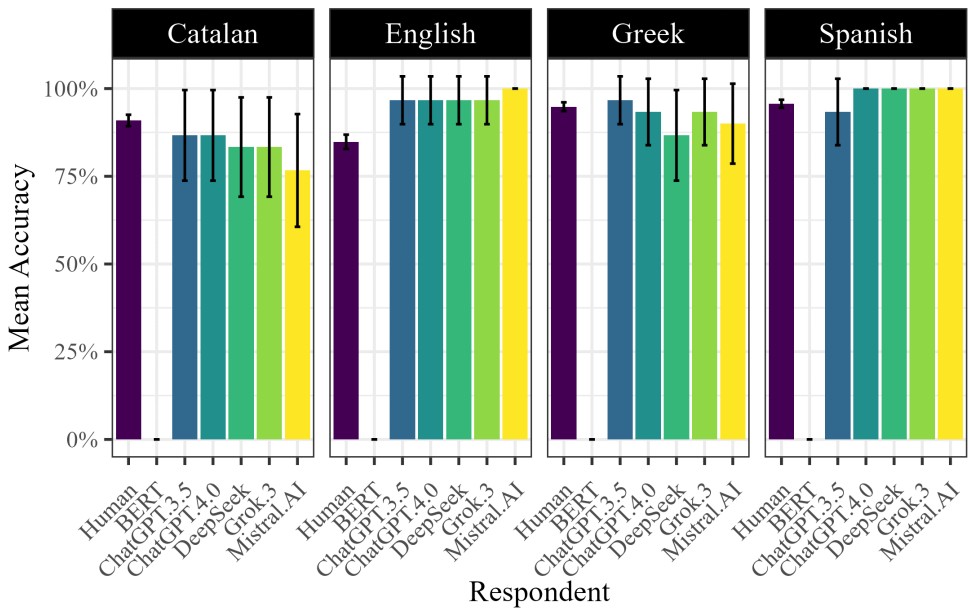

**Fig 3. Agent comparisons: Accuracy scores.**

averaged 95.6%. In English, models also surpassed humans, with Mistral at 100% compared to the human mean of 84.8%. In contrast, in Greek and Catalan, models did not (generally) outperform humans.

In order to determine if LLMs performed statistically different from humans, a mixed-effects logistic regression model (glmer) was also run with Accuracy (correct vs. incorrect) as the dependent variable and Agent (human vs. LLM) as the independent variable. Participants and items were added as crossed random factors, allowing for random intercepts. Given some of the variability present across the tested LLMs, three different glmers were run with. i) humans vs all LLMs, ii) humans vs. all LLMs except BERT, and iii) humans vs. only the best performing LLM (ChatGPT-4). Table 5 shows that Agent was only a significant predictor of accuracy when BERT was included in the analysis ($\chi^2 = 5.917$, $p = 0.015$). When BERT was excluded, Agent was not significant ($\chi^2 = 0.085$, $p = 0.77$). Similarly, when the best performing model was compared to humans, it yielded no significant effect of Agent ($\chi^2 = 0.002$, $p = 0.96$).

Despite LLM performing on par with humans when their scores were averaged across languages, we wanted to determine if LLM vs. human performance was alike within each language. To test this, another set of glmers were run, but this time including Language and Agent as fixed factors: in the first model they were included as two main effects and in the second model as an interaction. Model selection was performed by comparing the logLikehoods of the models using the anova()-function. When all LLMs (i.e., also BERT) were included in the data analysis, results show a significant main effect of both Agent ($\chi^2 = 5.837$, $p = 0.016$) and Language ($\chi^2 = 31.846$, $p < 0.001$); with humans performing significantly

**Table 5. Summary of mixed-effects regression model with Agent as main effect.**

| Comparison | β | SE | z | p |
|---|---|---|---|---|
| Humans vs. LLMs (excl. BERT) | −0.1813 | 0.6231 | −0.2910 | 0.7711 |
| Humans vs. ChatGPT-4 | −0.0625 | 1.2927 | −0.0484 | 0.9614 |
| Humans vs. All LLMs | −1.6087 | 0.6613 | −2.4325 | 0.015* |

better than models across all languages (β = 1.55, SE = 0.641, z = 2.416, p = 0.016). When BERT was excluded from the data analysis, a significant interaction between Agent and Language was found ($\chi^2$ = 28.323, p < 0.001). Post-hoc comparisons using the emmeans()-function with Tukey adjustment show that Agent did not play a significant role across the tested languages with the exception of English, where humans performed statistically worse than LLMs (Table 6).

Error analyses provide further insight into language-specific patterns. In English, both humans and models occasionally overgeneralized irregular plural forms (e.g., *sungi* for *sungus*). Humans additionally produced real-word substitutions (e.g., *fomputer→computer*), typos (e.g., *macumber* for *mucumbers*) and inserted irrelevant words such as "more" or "yes"; these types of errors were not observed in models. In Greek, humans and models alike struggled with particular stems (e.g., pluralizing *λίγρης* as *λίγρες* or *λίγρηδες*), though humans produced typos absent in models. In Catalan, both agents often failed to apply orthographic adjustments, yielding forms such as *fronjes* instead of *fronges*. In Spanish, errors were rare, though some humans inserted extraneous words (e.g., *sí* or *no*), while ChatGPT-3.5 occasionally failed to pluralize. Overall, models tended to mirror human error patterns, though human responses contained a wider range of idiosyncratic deviations, including especially irrelevant insertions. Table 7 provides examples of correct and incorrect responses produced by both agents. A full breakdown of error types is available in the results files of the "Humans" and "Models" folders of the OSF repository.

**Table 6. Results of the post-hoc test (excluding BERT) on the effect of Agent (human – LLM) per language. Values are given in log-odds ratio. P-value adjustment was conducted with the Tukey method. Asterisks indicate statistically significant contrasts.**

| Human - LLMs | β | SE | z | p |
|---|---|---|---|---|
| English | −1.948 | 0.736 | −2.646 | 0.0081* |
| Spanish | −0.975 | 0.884 | −1.103 | 0.2701 |
| Catalan | 0.986 | 0.593 | 1.663 | 0.0964 |
| Greek | 0.708 | 0.635 | 1.114 | 0.2652 |

**Table 7. Sample of correct and incorrect responses from humans and LLMs.**

| Language | Given example in singular | Humans' correct answers | Humans' responses with anomalies & failures | LLMs' correct answers | LLM(s') responses with anomalies & failures |
|---|---|---|---|---|---|
| English | nandy | nandies | nandys | nandies | child |
| English | sungus | sunguses | sungi/sungu/sungus's | sunguses | sungi/sung |
| English | luty | luties | lutties/lutys/luty's | luties | lutys/rain |
| English | watellite | watellites | watellies/wattelites/satellites | watellites | tree |
| Catalan | famí | famins | Famís/famils/ famíssos | – | famís/ famílics/ més |
| Catalan | madira | madires | – | madires | més |
| Catalan | fronja | fronges | fronja/franges/fronjes/frongues | – | fronjes/ més |
| Catalan | claça | claces | classes/ claçes/calces | – | claçes/claques/ més |
| Spanish | zoche | zoches | zoche/zocheas | zoches | árbol |
| Spanish | damino | Daminos | daños/dominos | daminos | hombre |
| Spanish | troplema | troplemas | – | troplemas | árbol |
| Spanish | danta | dantas | dantes/dardas/frutos | dantas | niño |
| Greek | φοντίκι (fodiki) | φοντίκια (fodikia) | ροντίκια(rodikia)/ ποντίκια (podikia) | φοντίκια (fodikia) | Φωντίκια (fodikia misspelled) |
| Greek | λύννεφο (linnefo) | λύννεφα (linnefa) | λύννεφο (linnefo) | λύννεφα (linnefa) | λύνεφα (linefa)/ βιβλίο (vivlio) |
| Greek | τίλος(τίλοι) | τίλοι (tili) | – | τίλοι (tili) | ακόμη (akomi) |
| Greek | λαρέκλα(larekla) | λαρέκλες(larekles) | – | λαρέκλες(larekles) | άλλο(allo) |

Taken together, these findings confirm that both humans and models perform well in generalizing morphological rules to novel words, with comparable overall accuracy. However, performance is modulated by language: as shown above English emerges as the most difficult system for humans, while Catalan proves most challenging for models.

In order to better understand the effect of Language on the performance of LLMs, another mixed-effects logistic regression model (glmer) was run on the LLMs data. As before, Accuracy (correct vs. incorrect) was coded as the dependent variable and Language (Catalan, English, Spanish, Greek) as the independent variable. Items and models were added as crossed random factors, allowing for random intercepts when possible. Given that BERT behaved significantly different from all the other LLMs, and that when including it in the data analysis the statistical models did not comply with some of required assumptions (e.g., within-groups deviations from uniformity), the rest of the paper will report the results of the data analysis excluding BERT. It must be highlighted that the main findings are not affected by whether this model is included or not, and any discrepancies will be reported (see the OSF repository, file Model_analyis.pdf for a comparison of including and excluding this model).

Results show a significant main effect of Language on accuracy ($\chi^2 = 26.517$, $p < 0.001$). A post-hoc analysis was conducted with the emmean()-function to better understand the differences across languages. Table 8 shows the results. As can be seen, LLMs performed significantly worse in Catalan compared to all the other tested languages.

The pattern observed is thus very different from the one obtained from humans (Table 4); a result which raises the question of which factor best determines the accuracy of LLMs cross-linguistically. As was mentioned in the introduction, two different hypotheses have been put forth in the literature to explain LLMs performance: the language's complexity and the language's community size. To determine which one is a better predictor, we run two additional mixed-effects logistic regressions with Accuracy as the dependent variable and Items and Model (if the model converged) as cross-random effects. In one of the models, the independent variable was the Complexity Score of the tested languages, and in the other the Community Size (z-scored). Results show a significant effect of Complexity Score ($\chi^2 = 5.678$, $p = 0.017$), and Community Size ($\chi^2 = 11.991$, $p = 0.0005$). Model selection is performed by comparing Akaike Information Criterion (AIC). The AIC was lower in the model with Community Size as a main effect (AIC = 277.858) than when Complexity Score was the main effect (AIC = 290.659). This suggests that, out of the two predictors, Community Size is the better one. The same conclusions are reached if BERT is included in the data analysis.

Lastly, we wanted to determine whether there is an interaction between the two predictors (Community Size and Complexity Score). The previous two models (the one with Community Size as a main effect and the one with Complexity Score as the main effect) were compared with a model in which both Community Size and Complexity Score were main effects (model 2), and with a model in which there was an interaction between Community Size and Complexity Score (model 3). The models were compared by means of the logLikelihood of the models using the anova()-function. Results showed that the best fitting model was the one with the interaction ($p < 0.001$). Importantly, this model shows that there is a significant interaction between the two main predictors ($\chi^2 = 5.837$, $p = 0.016$). To better understand the effect of the interaction, the predictions of the model were plotted using the ggeffects package [46] (Fig 4).

**Table 8. Results of the post-hoc test on LLMs (excluding BERT). Values are given in log-odds ratio. P-value adjustment was conducted with the Tukey method. Asterisks indicate statistically significant contrasts.**

| Contrast | ß | SE | z | p |
|---|---|---|---|---|
| Catalan – English | −2.250 | 0.584 | −3.855 | 0.0007** |
| Catalan – Greek | −0.989 | 0.408 | −2.426 | 0.0722 |
| Catalan – Spanish | −2.984 | 0.770 | −3.875 | 0.0006** |
| English – Greek | 1.261 | 0.613 | 2.057 | 0.1674 |
| English – Spanish | −0.734 | 0.890 | −0.826 | 0.8424 |
| Greek – Spanish | −1.995 | 0.792 | −2.520 | 0.0569 |

**Fig 4. Predictions of LLMs accuracy based on complexity score and Community Size.**

Overall, these results highlight the role not only of Complexity but also of Community Size in determining LLMs' accuracy in deriving plural regularizations patterns. The bigger the community size, the better the models are predicted to perform. Interestingly, if two languages have a similar number of speakers, the more complex language is predicted to outperform the other one.

## Discussion

This study examines how LLMs generalize morphological patterns across languages by addressing three main research questions: whether LLMs exhibit human-like behavior in the generalization of novel morphological forms (RQ1); whether linguistic complexity influences performance (RQ2); and whether accuracy is instead primarily driven by training data size and speaker community (RQ3).

In relation to RQ1, our findings show that LLMs performed remarkably similarly to humans. Excluding clear outliers such as BERT, whose performance was uniformly at floor across languages, mixed-effects analyses revealed no statistically significant differences between the two agents. Moreover, models matched human accuracy in three of the four languages, even outperforming significantly humans in English. These results indicate that LLMs are able to generalize morphological rules to unseen items with a level of reliability that is comparable to that of humans, replicating previous evidence contending that models can successfully reproduce certain aspects of human morphological reasoning, at least in constrained, rule-based tasks [19,22].

The main contribution of the present study concerns RQ2 and RQ3, which probe the relative influence of linguistic complexity and data availability on the models' performance. Our results reveal that both factors affected accuracy, but community size emerged as the stronger predictor. While linguistic complexity did lead to some variability in LLM accuracy – especially in morphologically rich languages like Catalan and Greek – model performance in languages with smaller speaker populations and consequent limited digital presence is systematically worse: LLMs were less accurate in Catalan and Greek, while English and Spanish, both supported by vast digital corpora, crucially achieved more consistent and higher accuracy.

One might object that these results could also reflect an effect of linguistic complexity, since Catalan and Greek are ranked highest also according to this metric. However, a closer examination of the results in Fig 3 reveals a clear

asymmetry: the ranking of model performance observed in our study (i.e., Spanish > English > Greek > Catalan) aligns more closely with the distribution of community size (English > Spanish > Greek > Catalan) than with that of linguistic complexity (English < Spanish < Catalan < Greek). Statistical analyses corroborate this pattern, indicating that although both factors contribute, community size is the strongest predictor of accuracy. Interestingly, moreover, the effect of linguistic complexity runs counter to conventional expectations found in the literature: when community sizes are comparable, our model predicts that greater complexity is in fact associated with higher accuracy.

A related and somewhat surprising observation concerns English, the least complex and most widely spoken language in our sample. Despite this advantage, it was not the best-performing language for either humans or models. Instead, Spanish consistently yielded the highest model accuracy, even though it is more morphologically complex and has a smaller speaker community.

This unexpected outcome —which potentially contradicts our main claim that community size better predicts model accuracy— deserves some clarification. In our study, the English task contained tokens that resemble words with irregular plural forms, thus complicating rule generalization for both humans and models. Spanish, by contrast, exhibits highly regular paradigms across all stimuli, which likely facilitated the models' consistent accuracy. Indeed, in English, both human participants and models alike occasionally extended familiar pluralization patterns inappropriately, producing forms such as *sungi* (target: *sunguses*) or *lutys* (target: *luties*). These analogical extensions mirror human tendencies in morphological generalizations and align with [47]'s observation that LLMs and humans both rely on structural analogy when confronted with novel linguistic material. The inherent irregularities of the English stimuli may thus be argued to contribute to lower accuracy scores.

Moreover, the comparatively weaker performance of English relative to Spanish may reflect broader typological differences: Germanic languages generally display greater morphological irregularities than Romance languages [48], which can hinder token- or pattern-based generalization in models. Similarly, Greek, despite its smaller speaker community, shows high accuracy, likely due to its morphological regularity. Together, these observations suggest that while data availability is the primary driver of LLM performance, internal morphological consistency may also moderate accuracy.

This nuanced conclusion refines existing claims in the literature. Whereas [22] identify linguistic complexity as the stronger predictor of model behavior, our results instead underscore the predominant importance of training resources, tempered by the interaction with morphological regularities. This becomes particularly evident when English performance is set aside. For instance, Greek which is the most complex language according to our metrics, did not occupy the lowest position; on the contrary, it systematically outperformed Catalan, which is relatively linguistically simpler. This finding is unexpected if grammatical complexity were the main determinant of model performance. Even more strikingly, Spanish and Catalan are ranked very closely in terms of linguistic complexity, yet their results diverge significantly, as shown in Table 8. This difference aligns with their relative representation in the training data: Spanish, far more digitally present, achieved markedly higher model accuracy than Catalan, which is poorly represented. Taken together, these findings challenge the assumption that linguistic complexity is a direct proxy for enhanced model performance. and instead foreground the decisive role of data resources.

Put differently the results support the conclusion that LLMs exhibit a high degree of resource sensitivity while remaining largely insensitive to linguistic complexity. Although these two factors undoubtedly interact, the evidence indicates that model performance is conditioned primarily by the quantity and representativeness of their training data, rather than by their sensitivity to the internal structural complexity of natural language systems. Models excel in languages with abundant and well-digitized corpora, but they do not exhibit systematic sensitivity to the morphological intricacies of those languages. This finding suggests that LLMs may rely on more mechanistic processes, such as tokenization, to parse language [3,7,8]. Such an interpretation aligns with research showing that sub-word tokenization methods like Byte Pair Encoding [49] privilege high-frequency morphological patterns and thereby benefit resource-rich languages like Spanish.

Turning briefly to human accuracy, several factors may help explain why performance did not reach ceiling levels. Some participants produced incorrect responses in the initial test items, likely due to a misinterpretation of the task instructions. For example, when prompted with "Now there are two ___?" several participants replied "*sí*" 'yes', which implied that they were evaluating the truth value of the sentence rather than filling in the gap. In other cases, responses that were semantically appropriate (i.e., that correctly performed the task at stake, namely pluralization) were coded as incorrect due to orthographic or prosodic inaccuracies, such as misspellings, typos, or misplaced stress. Such deviations, which were absent from the models' output, likely reflect human-specific performance factors such as attention lapses, cognitive fatigue, or time pressure.

Certain limitations of this study warrant consideration. First, although linguistic complexity scores were computed using data from a typologically diverse linguistic database, these scores encompass a broad range of linguistic features extending beyond morphology to syntax and semantics. While such holistic measures provide valuable insights into cross-linguistic diversity, they may not align precisely with the specific focus of this study. Given that the Wug Test primarily targets inflectional morphological processes, it would be more accurate to isolate and assess *morphological complexity* as a distinct construct. A morphology-focused metric would likely alter the relative scores and enable more precise cross-linguistic comparisons. Future research would therefore benefit from constructing dedicated indices of morphological complexity, which take into consideration factors such as paradigm size, rule regularity, allomorphy, and morphological transparency, to better contextualize both model and human performance.

A second limitation concerns the relatively small number of languages tested. Although the selected languages were chosen to ensure diversity in both training data size and structural complexity, this sample restricts the generalizability of the findings across the world's languages, especially those that are underrepresented. Including languages with radically different typological profiles (e.g., agglutinative, polysynthetic, or tonal systems) would offer a more comprehensive understanding of how LLMs handle morphological generalization under varying linguistic and data conditions.

A third limitation concerns the strength of the link we have established between size of the community and size of the training data. While there is a connection between the two —a small community that speaks a minoritized, non-official language will likely have less resources, and this scarcity of high-quality digitalized data will cast an upper limit on LLM performance—, taking community size as a proxy for volume of training data is neither that simple nor that straightforward. For example, Basque, a language isolate with no demonstrable relationship with any other language, has a community of about 800,000 people. Swahili, a Bantu language spoken across several countries, has a community of 150 million people. However, Basque has a much stronger digital presence than Swahili, with six times more Wikipedia articles in the former than in the latter [50] This suggests that while a link between community size and training data size exists, this is modulated by many geopolitical factors, such that the so-called 'low-resourced' languages are not a uniform category. This weakens any attempt to establish a direct relationship between 'low-resourced' vs. 'high-resourced' language and LLM performance.

A last limitation concerns prompt-related bias. Even subtle changes in wording can significantly alter LLM replies in linguistic tasks. It is possible that choosing different prompts or different nonce words would have led to different results. However, as our results agree with what has been reported in the literature through different prompts and task designs ([19,22]), we assess this possibility as relatively low.

## Conclusion

This study examined how LLMs extend morphological generalizations to novel words across languages differing in linguistic complexity and community size. The results revealed that while models can replicate human-like accuracy, their success is especially determined by the availability of training data rather than on grammatical complexity. Languages with broader community sizes and greater digital representation, such as Spanish and English, consistently yielded higher model accuracy than less represented ones like Greek and Catalan. At the same time, the regularity of a language's

morphological system can moderate performance, allowing even poorly represented languages to achieve higher accuracy when structural patterns are transparent and consistent. These findings suggest that while LLMs remain powerful pattern learners, their behavior reflects a distributional, rather than structural, grasp of language: one that mirrors human performance only in output but not necessarily in the underlying mechanisms.

## Acknowledgments

We would like to thank Sergi Balari and Elena Pagliarini for their valuable feedback on previous versions of this work. We are also grateful to M.Teresa Espinal for her help with the Catalan stimuli, and Olena Shcherbakova and Hedvig Skirgård for their assistance with the Grambank complexity scores. All remaining errors are our own.

## Author contributions

**Conceptualization:** Nikoleta Pantelidou, Evelina Leivada.

**Data curation:** Raquel Montero.

**Formal analysis:** Nikoleta Pantelidou, Raquel Montero.

**Methodology:** Nikoleta Pantelidou, Evelina Leivada.

**Writing – original draft:** Nikoleta Pantelidou, Paolo Morosi.

**Writing – review & editing:** Nikoleta Pantelidou, Evelina Leivada, Raquel Montero, Paolo Morosi.

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
