## [Decision Letter · Decision Letter 0]

15 Dec 2025

Dear Dr. Morosi,

Thank you for submitting your manuscript to PLOS ONE. After careful consideration, we feel that it has merit but does not fully meet PLOS ONE’s publication criteria as it currently stands. Therefore, we invite you to submit a revised version of the manuscript that addresses the points raised during the review process.

We look forward to receiving your revised manuscript.

Kind regards,

Wei Lun Wong

Academic Editor

PLOS One

Journal Requirements:

Reviewers' comments:

Reviewer's Responses to Questions

**Comments to the Author**

1. Is the manuscript technically sound, and do the data support the conclusions?

Reviewer #1: Partly

Reviewer #2: Yes

Reviewer #3: Yes

Reviewer #4: Yes

Reviewer #5: Yes

2. Has the statistical analysis been performed appropriately and rigorously?

Reviewer #1: No

Reviewer #2: Yes

Reviewer #3: Yes

Reviewer #4: Yes

Reviewer #5: No

3. Have the authors made all data underlying the findings in their manuscript fully available?

Reviewer #1: Yes

Reviewer #2: Yes

Reviewer #3: Yes

Reviewer #4: Yes

Reviewer #5: Yes

4. Is the manuscript presented in an intelligible fashion and written in standard English?

Reviewer #1: Yes

Reviewer #2: Yes

Reviewer #3: Yes

Reviewer #4: Yes

Reviewer #5: Yes

Reviewer #1: This paper examines how Large Language Models apply morphological rules to novel words using a multilingual version of the Wug Test. By comparing six models across four languages with human speakers, the study evaluates whether model performance reflects true linguistic competence or merely the amount of training data available. The findings suggest that data availability and community size, rather than linguistic complexity, primarily shape model accuracy.

- Please explain the motivation / scientific importance for comparing particularly community size and grammar complexity.

- In the abstract, please distinguish between, resource size and community size, as it creates confusion.

- Please add a model diagram in the methodology section.

- It is suggested to use more appropriate (specific) wording than "language-blind" as language could mean many things. At present it leads to confusion.

- Colors in Fig 2 are indistinguishable, please add visibly separate patterns over the bars.

- The study will benefit from if authors separately test the generative LLMs from reasoning/thinking LLMs.

- "English, the least complex of the four languages we tested, was not the best-performing language for either humans or models. Instead, Spanish consistently yielded the highest model accuracy, despite its greater linguistic complexity." This authors' conclusion is in contrast to their main claim in the paper. Authors need to investigate as to what is the actual reason behind this, when English is simple, has larger community, and more resources?

- Similarly authors need to investigate that why: "At the same time, Greek, which is the most complex language according to our metrics, did not occupy the lowest position; on the contrary, it systematically outperformed Catalan, which is relatively linguistically simpler."

- Authors should perform statistical significance test to verify the relevance of these claims.

- Please explain the compute time and resources that were invested to conduct the study.

- For the Wug Test, give multiple examples in the results showing how different models performed and how humans performed. Also add the cases where reported anomalies were seen. Like the failure cases and the unexpected anomalistic cases.

- The length of the paper appears to be rather short, more emphasis is given on literature review, however the presentation can be improved. It is suggested to make a chronological table of all the studies performed on this topic and list down their conclusions/ key-findings, experimental setup, datasets used.

- Furthermore, the results section need to be strengthen, showing multiple examples/results.

- Please explain why and how accuracy is selected as the metric of choice. In many cases it is not the correct measure of the performance, also add the AUC, recall, sensitivity, f1-score, precision scores.

Overall, the manuscript appears to make useful contribution, but further justifications are required.

Reviewer #2: Dear Author,

The manuscript presents a clear and well-designed study examining how Large Language Models generalize morphological rules across four languages using a multilingual Wug Test. The research question is timely, and the methodology—especially the construction of nonce stimuli, the balanced design, and the use of GLMMs—is appropriate and transparent. Ethical approval, participant recruitment, and data availability are all thoroughly documented.

The results are clearly presented, and the interpretation is reasonable, particularly the conclusion that model performance aligns more with community size and data exposure than with structural complexity. Some claims, however, would benefit from slightly more cautious wording. The limitations section could also briefly address potential prompt-related biases when interacting with different models.

Overall, this is a strong and relevant contribution. With minor clarifications and small stylistic adjustments, the manuscript would be suitable for publication in PLOS ONE.

Reviewer #3: Thank you for the opportunity to review this interesting and timely manuscript. The study raises valuable questions; however, several areas would benefit from clarification, refinement, and further detail to strengthen the overall contribution. My detailed comments are as follows:

* Lines 252–254: This paragraph does not appear necessary and could be removed without affecting the clarity or structure of the manuscript.

* Wug test materials: You mention 30 items; however, the file provided (“Humans: Task and stimulus: novel words.xlsx”) shows 15 two-syllable and 15 three-syllable words. It is important to clarify this breakdown in the manuscript and explicitly indicate that the full list is available in the supplementary materials.

* Selection criteria for nonce words: Please provide more information on how the nonce words were selected. Clarifying the linguistic criteria used would improve methodological transparency.

* Targeted morphological processes: Briefly state which morphological processes were targeted in the Wug test (e.g., inflectional morphology). Explain the motivation behind selecting these particular processes and whether this selection is supported by prior research.

* Lines 296–297: The exclusion criteria include cognitive, neurological, hearing, or speech-related impairments. Please comment on whether these criteria could have influenced the results. If these factors are not expected to affect outcomes, clarify the rationale for including them.

* Mode and medium of testing: The manuscript should specify whether the test was administered in written form, spoken form, or both. If spoken, please indicate whether any recording requirements or controls were implemented for human participants.

* Line 339: The link provided earlier is repeated here; the duplication is unnecessary.

* Table 3: It is unclear why Catalan was not reported in this table. Please clarify or revise accordingly. Additionally, the inclusion of “(Intercept)” requires explanation, and it would be helpful to comment on this table in the main text.

* Language quality: A careful proofreading pass is needed to address minor issues with capitalization and punctuation.

* Line 394: The phrase “errors not observed in models” requires clarification. Does this refer to all errors, or only the specific error types discussed?

* Lines 401–402: When referencing the full breakdown of error types available in the OSF repository, please specify the file name to guide readers.

* Lines 476–477: The statement that “Germanic languages display higher levels of morphological irregularity than Romance languages” introduces a comparative rationale. If this comparison is central to the study, please present it earlier in the Methods section. It would also be helpful to explain the rationale for selecting these four languages and what distinguishes them.

* Choice of languages: More broadly, please articulate the logic behind selecting these four languages. What features or contrasts make them suitable for comparison in this context?

* Line 497: Consider rephrasing the sentence beginning with “Before concluding, it is important to acknowledge the limitations…” to avoid a dialogic construction.

* Statistical Analysis: The statistical framework chosen for this study is appropriate for the data structure. Using generalized linear mixed models (GLMMs) to model binary accuracy is sound, and treating test items as random effects is a clear strength. However, several aspects of the analysis would benefit from refinement to ensure full rigor and transparency. Most importantly, the models for human data do not include participants as random effects, despite repeated observations, which may underestimate variance and inflate the apparent significance of fixed effects. Additionally, the structure of the combined human–model analysis is under-specified, particularly regarding whether an Agent × Language interaction was included, and the reporting relies primarily on p-values without providing effect sizes or diagnostic checks. Finally, the conclusions about linguistic complexity versus community size extend beyond what is directly supported by the statistical tests presented. Addressing these points would substantially strengthen the analytical robustness of the manuscript.

* Use of tools: Please ensure that all tools, platforms, or software referenced in the Methods and Results sections also appear in the References list.

* References: The reference list contains inconsistencies in formatting. Please revise to ensure adherence to the journal’s required style.

Overall, the study presents promising results, and addressing the points above will help enhance the clarity, rigor, and completeness of the manuscript.

Reviewer #4: The manuscript presents a technically sound and methodologically rigorous study that clearly supports its conclusions. The experimental design, which is a controlled multilingual Wug Test administered to both human participants and six LLMs, is appropriate for examining morphological generalization, and the authors justify their choice of languages, stimuli construction, and procedures in detail. The statistical analyses are correctly executed and suitable for the research questions. Accuracy is modeled with GLMMs including random effects for items, model comparisons are performed through likelihood ratio tests, and estimated marginal means are used to interpret cross-linguistic differences. These methods are transparent, replicable, and provide robust support for the claims made. Data availability fully complies with PLOS ONE policies, with all raw data, code, and stimuli openly accessible on OSF without restrictions. The manuscript itself is clearly written, logically structured, and expressed in standard academic English. While minor phrasing issues appear occasionally, they do not impede comprehension or interpretation of the findings. Overall, the authors deliver a well-executed study that contributes meaningfully to ongoing debates about LLM linguistic competence and presents results in a way that is both empirically grounded and accessible to a broad readership.

Room for improvement:

Although the manuscript is well executed, several weaknesses limit the strength of its conclusions. First, the operationalization of “linguistic complexity” is not fully aligned with the study’s goals. The authors rely on global Grambank fusion and informativity scores (e.g., lines 147–166), but these metrics incorporate many grammatical domains irrelevant to nominal morphology. This weakens claims about the relationship between morphological complexity and model accuracy, especially when the authors later acknowledge (lines 499–507) that a morphology-specific measure would be more appropriate. Second, the English stimuli appear to contain several nonce forms that unintentionally resemble irregular plural patterns (e.g., sungus, lutie), which the authors note may have misled both humans and models (lines 467–474). Because these irregularity-triggering forms are not systematically described or quantified, it is difficult to assess whether English’s lower accuracy reflects linguistic complexity, task design artifacts, or item-specific biases. Finally, some claims in the discussion appear stronger than the data justify. For example, the conclusion that models are “language-blind” and guided primarily by resource availability (lines 453–457) may overgeneralize from only four languages, two of which are typologically similar (Catalan and Spanish). These weaknesses should be addressed to strengthen the study’s empirical and theoretical claims.

Reviewer #5: Good paper, interesting idea. I like the Wug Test setup across languages. The main finding is solid and worth publishing. But there are some pretty big problems to fix first.

• Stats need a major clean-up. The numbers in the paper don't add up enough for me to really trust them yet.

• That Table 3 is confusing. What's it comparing to? I need to see the full stats table.

• Saying results aren't "significant" with p=.055 and .092 is kind of shaky. That's really close. You can't just say "they're the same" and move on. Talk about what those borderline numbers might mean.

• Kicking BERT out because it got 0% feels like cheating. If it's that bad, that's actually a cool finding! Either put it back in and explain why it's so different, or give a better reason upfront for why it doesn't count.

• "Community Size" = Training Data? Not so fast. Your whole argument rests on this, but it's a huge assumption. Just because more people speak Spanish doesn't mean GPT saw exactly that much more Spanish text. The internet is weird. You need to either defend this link way better with evidence, or tone down your conclusions a lot and call this a major guess you had to make.

• You're over-selling it. Calling LLMs "language-blind" in the title is too much. Your own data shows they kind of notice if a language is regular (like Spanish). Tone it down. Also, be careful saying they only have "superficial" competence: that's a philosophy paper. Your experiment just shows they're good at this specific pattern-matching task, and their skill depends on how much stuff they've read.

• Tell us exactly what you typed into ChatGPT. The prompt matters.

• Figure 2 is a mess of lines. Make it simpler.

• Fix the references. Some are missing info.

• Don't blame "hard test items" for English being tough. Just stick with the "Germanic languages are irregular" explanation, which is better.

The core idea is cool and the paper should eventually be published. But you got to fix the stats, be more honest about the "community size" guess, and don't claim more than you actually proved. Do that, and you'll have a much stronger paper.

**Do you want your identity to be public for this peer review?** For information about this choice, including consent withdrawal, please see our Privacy Policy

Reviewer #1: No

Reviewer #2: No

Reviewer #3: **Yes:**  Nada AlJamal

Reviewer #4: **Yes:**  Parisa Etemadfar

Reviewer #5: **Yes:**  EBA TERESA GAROMA

---

## [Author Response · Author response to Decision Letter 1]

20 Jan 2026

Dear Reviewers and editor,

We greatly appreciate the comments, questions and suggestions from the five Reviewers, as they have significantly contributed to the improvement of the revised paper we are now resubmitting.

Please see below the responses to the specific comments made by the five reviewers. You can also access these responses in the file "Response to Reviewers" we have uploaded with our resubmission.

Sincerely,

Dr. Paolo Morosi on behalf of all the authors

We thank the Editor for the helpful reviews they secured for us and the five Reviewers for their comments, which have contributed to strengthening our work. Below, we respond to each comment and list the changes we made in the revised version of the manuscript.

Reviewer #1

This paper examines how Large Language Models apply morphological rules to novel words using a multilingual version of the Wug Test. By comparing six models across four languages with human speakers, the study evaluates whether model performance reflects true linguistic competence or merely the amount of training data available. The findings suggest that data availability and community size, rather than linguistic complexity, primarily shape model accuracy.

- Please explain the motivation / scientific importance for comparing particularly community size and grammar complexity.

We thank the Reviewer for their assessment and for the very helpful comments which we address below, linking them to the revised manuscript.

- Please explain the motivation / scientific importance for comparing particularly community size and grammar complexity.

In the revised manuscript, the motivation has been added on pp. 6-7, lines 130-133.

- In the abstract, please distinguish between, resource size and community size, as it creates confusion.

The abstract has been rephrased in order to make the connection between the two notions, community size and volume of training data, clearer. To some degree, the latter is conditioned by the former. For example, a small community that speaks a minoritized language is unlikely to have a strong digital presence. This explains why the performance of LLMs has been found to be better in big, standard, official languages. On pp. 29 (lines 643-654) of the revised manuscript, we have also added discussion on low-resourced vs. high-resourced languages in order to better justify the link we put forth between training data/resource size and community size.

- Please add a model diagram in the methodology section.

A model diagram has been added as Fig 1 on pp. 15.

- It is suggested to use more appropriate (specific) wording than "language-blind" as language could mean many things. At present it leads to confusion.

The title has been changed.

- Colors in Fig 2 are indistinguishable, please add visibly separate patterns over the bars.

We have changed the colors and and also the presentation of Figure 3. Now the models appear on the x-axis and the languages are separated across different panels.

- The study will benefit from if authors separately test the generative LLMs from reasoning/thinking LLMs.

In the Agent_analysis file in the OSF repository (https://osf.io/4z5n6/, go to: Code > Humans_and_Models_Wug_Test), in Sections 3.4 and Section 3.5 we now also include this analysis comparing reasoning and general LLMs with humans. While only general purpose models are the ones that come as different from humans, this is due to the effect of BERT; once this model is removed (section 3.5) neither reasoning nor non-reasoning are statistically different from humans.

- "English, the least complex of the four languages we tested, was not the best-performing language for either humans or models. Instead, Spanish consistently yielded the highest model accuracy, despite its greater linguistic complexity." This authors' conclusion is in contrast to their main claim in the paper. Authors need to investigate as to what is the actual reason behind this, -when English is simple, has larger community, and more resources?

We thank the reviewer for raising this important point, which we addressed in the revised manuscript. Our new statistical analysis indeed confirms that both community size and linguistic complexity are significant predictors of model accuracy, and that community size is in fact the stronger predictor. These results align with the core claim of our manuscript.

However, the reviewer is right that, given that English is linguistically simpler and has the largest speaker community, it not being the best-performing language comes as unexpected. Nonetheless, it is important to note that the two factors, community size and linguistic complexity, interact in ways that are difficult to fully disentangle with currently available metrics. As discussed in the revised version of the manuscript, we relied on a global complexity score, following previous work. Although Spanish comes out as more complex than English overall, specific subcomponents of complexity (e.g., morphological complexity in plural formation in the case at hand), may differ across languages. It is conceivable, for instance, that English plural morphology presents greater irregularity in the specific Wug-test domain we examined, which could depress model performance relative to Spanish. This interpretation, however, is only speculative, given the lack of fine-grained complexity metrics.

Importantly, however, this line of reasoning would also predict Spanish and Catalan to pattern similarly, as both languages share comparable morphological properties, especially in plural formation. Yet our data show that Catalan performed substantially worse than both English and Spanish. This discrepancy is one of the reasons that led us to hypothesize that community size – and thus representation in the training data – is likely to be a better predictor of model accuracy. Our new statistical analysis supports this interpretation.

A further factor that may have disproportionately affected English performance relates to broader typological tendencies. As discussed in the manuscript, Germanic languages have been reported to exhibit higher levels of morphological irregularity than Romance languages, which can hinder token- or pattern-based generalization in models. These irregularities are also reflected in our prompt design: we deliberately used pseudo-words modeled on irregular nouns (e.g., sungus from fungus) to probe the limits of the models’ generalizations. Together, these factors may have increased error rates specifically in English, as models—like humans—sometimes produced the irregular plural form rather than the target generalization. This effect would be less pronounced in Spanish or Catalan, where irregular plural patterns are less frequent.

In sum, although we can identify plausible contributing factors, we agree that a complete explanation for the relative performance of English cannot be provided with certainty at present. We have now addressed this point more explicitly in the revised manuscript and noted it as an avenue for future research.

- Similarly authors need to investigate that why: "At the same time, Greek, which is the most complex language according to our metrics, did not occupy the lowest position; on the contrary, it systematically outperformed Catalan, which is relatively linguistically simpler."

We thank the reviewer for raising this point. As noted in our response to the previous comment, disentangling the contributions of linguistic complexity and community size is challenging, given that both factors are significant predictors of model accuracy in our new statistical analysis. However, the specific pattern highlighted here, namely Greek outperforming Catalan despite being the more complex language, does not contradict our main claim. In fact, it supports it: if linguistic complexity were the primary determinant of performance (i.e. the less complex a language is, the better LLMs perform at it), Catalan should have ranked higher than Greek. The opposite outcome suggests that community size and, more broadly, digital representation plays a more decisive role. Greek has a substantially larger speaker community than Catalan. Therefore, this asymmetry provides a plausible explanation for why Greek systematically outperformed Catalan in our results. We have clarified this point in the revised manuscript on p. 27, lines 585-592.

- Authors should perform statistical significance test to verify the relevance of these claims.

Table 8 now includes the pair-wise comparisons across the different languages in the LLMs performance. In the model, Catalan is statistically different from English and Spanish; there are two border-line cases (i.e. Greek-Spanish and Catalan-Greek), but they do not reach significance.

- Please explain the compute time and resources that were invested to conduct the study.

3 paragraphs have been added in the Participants section in lines 320-340.

- For the Wug Test, give multiple examples in the results showing how different models performed and how humans performed. Also add the cases where reported anomalies were seen. Like the failure cases and the unexpected anomalistic cases.

In the Results and analysis section, p. 26, Table 7 has been added, which includes examples across all languages with the target responses as well as with the anomalies and failures by both humans and LLMs.

- The length of the paper appears to be rather short, more emphasis is given on literature review, however the presentation can be improved. It is suggested to make a chronological table of all the studies performed on this topic and list down their conclusions/ key-findings, experimental setup, datasets used.

The Introduction section was updated with Table 1, including the relevant studies that directly relate to ours.

- Furthermore, the results section need to be strengthen, showing multiple examples/results.

In the Results section, p. 26, Table 7 has been added, which includes examples across all languages with the target responses as well as with the anomalies and failures by both humans and models.

- Please explain why and how accuracy is selected as the metric of choice. In many cases it is not the correct measure of the performance, also add the AUC, recall, sensitivity, f1-score, precision scores.

The Wug Test taps into the ability to generalize rules to novel words that have not been encountered before. We do not measure precision or recall in this task. In line with previous literature, we coded Accuracy in this linguistic task as target/accurate (1) or non-target/inaccurate (0). AUC-ROC curves or F1 score are used with heavily imbalanced datasets, when traditional metrics like accuracy can be misleading, which is not the case here.

Overall, the manuscript appears to make useful contribution, but further justifications are required.

We thank once again the Reviewer for their very helpful feedback. We hope that we have provided the justifications required, but we remain available to further work any point the Reviewer may deem unclear.

Reviewer #2

The manuscript presents a clear and well-designed study examining how Large Language Models generalize morphological rules across four languages using a multilingual Wug Test. The research question is timely, and the methodology—especially the construction of nonce stimuli, the balanced design, and the use of GLMMs—is appropriate and transparent. Ethical approval, participant recruitment, and data availability are all thoroughly documented.

The results are clearly presented, and the interpretation is reasonable, particularly the conclusion that model performance aligns more with community size and data exposure than with structural complexity.

We thank the Reviewer for their positive assessment.

Some claims, however, would benefit from slightly more cautious wording. The limitations section could also briefly address potential prompt-related biases when interacting with different models.

We agree with the Reviewer. Various claims have been reworked in favor of more cautious wording, and especially the limitations section has been significancy expanded, also discussing the role of prompt-related biases (pp. 30, lines 656-660)

Reviewer #3

Thank you for the opportunity to review this interesting and timely manuscript. The study raises valuable questions; however, several areas would benefit from clarification, refinement, and further detail to strengthen the overall contribution. My detailed comments are as follows:

We thank the Reviewer for their helpful and detailed comments. We address each one of the below.

* Lines 252–254: This paragraph does not appear necessary and could be removed without affecting the clarity or structure of the manuscript.

We agree with the Reviewer. This paragraph has been removed from the revised manuscript.

* Wug test materials: You mention 30 items; however, the file provided (“Humans: Task and stimulus: novel words.xlsx”) shows 15 two-syllable and 15 three-syllable words. It is important to clarify this breakdown in the manuscript and explicitly indicate that the full list is available in the supplementary materials.

In the revised version, both these changes have been implemented (pp. 12-13).

* Selection criteria for nonce words: Please provide more information on how the nonce words were selected. Clarifying the linguistic criteria used would improve methodological transparency.

In the revised version, this information is given on p. 13, lines 289-291.

* Targeted morphological processes: Briefly state which morphological processes were targeted in the Wug test (e.g., inflectional morphology). Explain the motivation behind selecting these particular processes and whether this selection is supported by prior research.

In the revised version, this information has been added on p. 12, lines 263-271.

* Lines 296–297: The exclusion criteria include cognitive, neurological, hearing, or speech-related impairments. Please comment on whether these criteria could have influenced the results. If these factors are not expected to affect outcomes, clarify the rationale for including them.

We did not have any concrete expectations about the results, but these are typical exclusion criteria in studies that aim to establish neurotypical human baselines.

* Mode and medium of testing: The manuscript should specify whether the test was administered in written form, spoken form, or both. If spoken, please indicate whether any recording requirements or controls were implemented for human participants.

This information has been added on p. 13.

* Line 339: The link provided earlier is repeated here; the duplication is unnecessary.

The duplication has been removed.

* Table 3: It is unclear why Catalan was not reported in this table. Please clarify or revise accordingly. Additionally, the inclusion of “(Intercept)” requires explanation, and it would be helpful to comment on this table in the main text.

Table 3 has become now Table 4 and has been updated with the pairwise comparisons of the post-hoc test to show all the comparisions across all the languages. This was calculated using the emmeans()-function with Tukey adjustment.

* Language quality: A careful proofreading pass is needed to address minor issues with capitalization and punctuation.

We thank the reviewer for this recommendation. We have carefully proofread the revised manuscript and corrected minor issues with capitalization and punctuation.

* Line 394: The phrase “errors not observed in models” requires clarification. Does this refer to all errors, or only the specific error types discussed?

We thank the reviewer for this observation. We have revised the sentence to clarify that we were referring only to the specific error types discussed in the preceding sentence, not to all possible errors.

* Lines 401–402: When referencing the full breakdown of error types available in the OSF repository, please specify the file name to guide readers.

In lines 453-455 of the Results section, the file and folder names have been specified.

* Lines 476–477: The statement that “Germanic l

---

## [Decision Letter · Decision Letter 1]

2 Feb 2026

Community size rather than grammatical complexity better predicts Large Language Model accuracy in a novel Wug Test

PONE-D-25-55707R1

Dear Dr. Morosi,

We’re pleased to inform you that your manuscript has been judged scientifically suitable for publication and will be formally accepted for publication once it meets all outstanding technical requirements.

Kind regards,

Wei Lun Wong

Academic Editor

PLOS One

Additional Editor Comments (optional):

Reviewers' comments:

Reviewer's Responses to Questions

**Comments to the Author**

Reviewer #2: All comments have been addressed

Reviewer #3: All comments have been addressed

Reviewer #5: All comments have been addressed

2. Is the manuscript technically sound, and do the data support the conclusions?

Reviewer #2: Yes

Reviewer #3: Yes

Reviewer #5: Yes

3. Has the statistical analysis been performed appropriately and rigorously?

Reviewer #2: Yes

Reviewer #3: Yes

Reviewer #5: Yes

4. Have the authors made all data underlying the findings in their manuscript fully available?

Reviewer #2: Yes

Reviewer #3: Yes

Reviewer #5: Yes

5. Is the manuscript presented in an intelligible fashion and written in standard English?

Reviewer #2: (No Response)

Reviewer #3: Yes

Reviewer #5: Yes

Reviewer #2: he author has appropriately implemented all the required revisions. The revised manuscript shows clear improvements and meets the journal's scientific publication standards, with no major remaining concerns."

Reviewer #3: (No Response)

Reviewer #5: Review points properly addressed

• Motivation for comparing community size and grammatical complexity: The authors have added a clear explanation of the motivation in the Introduction (pp. 6-7). The connection between community size, digital representation, and training data availability is now better articulated, strengthening the rationale for the study.

• Clarification of “community size” vs. “resource size” in the Abstract: The abstract has been rephrased to clarify the relationship between community size and training data volume. The revised version is clearer and avoids potential confusion.

• Addition of a model diagram: A clear experimental design diagram (Fig. 1) has been added to the Methodology section, which improves the readability and reproducibility of the study.

• Removal of “language-blind” from the title; The title has been changed to a more precise and less overreaching formulation, which aligns better with the empirical findings.

• Improvement of Figure 2/3 readability: The revised Figure 3 now uses clearer color distinctions and a more intuitive layout (models on the x-axis, languages in separate panels), addressing my earlier concern about visual clarity.

• Additional analysis separating generative and reasoning LLMs: The authors conducted additional analyses comparing reasoning and non-reasoning models, which are now available in the OSF repository. This adds depth to the results and addresses my suggestion for further model-type comparisons.

• Statistical significance testing and post-hoc comparisons: The authors have rerun their statistical models with improved specifications (including participant random effects) and provided post-hoc comparisons in Table 8. The reporting is now more transparent and rigorous.

• Strengthened Introduction and Results sections: The Introduction now includes Table 1 summarizing relevant prior work, and the Results section is more detailed with added tables and examples, addressing my concern about the paper’s initial brevity.

Remaining Minor Suggestions

While the authors have done an excellent job revising the manuscript, a few minor points could still be polished:

• Reference formatting: Although the authors state that references have been corrected, I noticed a few inconsistencies in formatting (e.g., capitalization, use of “et al.”, DOI presentation). A final careful pass to ensure adherence to PLOS ONE style is recommended.

• Clarity in limitations: The limitation regarding the link between community size and training data is well-acknowledged. However, the authors might briefly suggest how future work could better operationalize this relationship (e.g., using corpus size estimates rather than speaker counts).

**Do you want your identity to be public for this peer review?** For information about this choice, including consent withdrawal, please see our Privacy Policy

Reviewer #2: **Yes:**  Dr. Neamah Dahash Farhan Professor in University of Baghdad / College of Islamic Sciences -Iraq

Reviewer #3: **Yes:**  Nada AlJamal

Reviewer #5: **Yes:**  EBA TERESA GAROMA

---

## [Editor Report · Acceptance letter]

PONE-D-25-55707R1

PLOS One

Dear Dr. Morosi,

I'm pleased to inform you that your manuscript has been deemed suitable for publication in PLOS One. Congratulations! Your manuscript is now being handed over to our production team.

Kind regards,

on behalf of

Dr. Wei Lun Wong

Academic Editor

PLOS One